# Early Re-Exploration versus Conservative Management for Postoperative Bleeding in Stable Patients after Coronary Artery Bypass Grafting: A Propensity Matched Study

**DOI:** 10.3390/jcm12093327

**Published:** 2023-05-07

**Authors:** Cristiano Spadaccio, David Rose, Antonio Nenna, Rebecca Taylor, Mohamad Nidal Bittar

**Affiliations:** 1Cardiothoracic Surgery, Lancashire Cardiac Center, Blackpool Victoria Hospital, Blackpool FY3 8NR, UK; mr.rose@nhs.net (D.R.);; 2Cardiovascular Surgery, Università Campus Bio-Medico di Roma, 00128 Rome, Italy; a.nenna@policlinicocampus.it; 3Research and Development, Blackpool Teaching Hospitals, Blackpool FY3 8NR, UK

**Keywords:** coronary artery bypass graft, bleeding, blood products, transfusion, complications

## Abstract

Background: Postoperative bleeding requiring re-exploration in cardiac surgery has been associated with complications impacting short-term outcomes and perioperative survival. Many aspects of decision-making for re-exploration still remain controversial, especially in hemodynamically stable patients with significant but not acutely cumulating chest drain output. We investigated the impact of re-exploratory surgery on short-term outcomes in a “borderline population” of CABG patients who experienced significant non-acute bleeding, but that were not in critically hemodynamic unstable conditions. Methods: A prospectively collected database of 8287 patients undergoing primary isolated elective CABG was retrospectively interrogated. A population of hemodynamically stable patients experiencing significant non-acute or rapidly cumulating bleeding (>1000 mL of blood loss in 12 h, <200 mL per hour in the first 5 h) with normal platelet and coagulation tests was identified (N = 1642). Patients belonging to this group were re-explored (N = 252) or treated conservatively (N = 1390) based on the decision of the consultant surgeon. Clinical outcomes according to the decision-making strategy were compared using a propensity score matching (PSM) approach. Results: After PSM, reoperated patients exhibited significantly higher overall blood product consumption (88.4% vs. 52.6% for red packed cells, *p* = 0.001). The reoperated group experienced higher rates of respiratory complications (odds ratio 5.8 [4.29–7.86] with *p* = 0.001 for prolonged ventilation), prolonged stay in intensive care unit (coefficient 1.66 [0.64–2.67] with *p* = 0.001) and overall length of stay in hospital (coefficient 2.16 [0.42–3.91] with *p* = 0.015) when compared to conservative management. Reoperated patients had significantly increased risk of multiorgan failure (odds ratio 4.59 [1.37–15.42] with *p* = 0.014) and a trend towards increased perioperative mortality (odds ratio 3.12 [1.08–8.99] with *p* = 0.035). Conclusions: Conservative management in hemodynamically stable patients experiencing significant but non-critical or emergency bleeding might be a safe and viable option and might be advantageous in terms of reduction of postoperative morbidities and hospital stay.

## 1. Introduction

Postoperative bleeding requiring re-exploration after cardiac surgery has been reported with incidences ranging between 2 and 13% [1,2,3] and carries a significant burden in terms of mortality, complications and resource expenditure [4]. Surgical re-exploration has been associated with end-organ damage due to extensive blood product transfusion and development of acute renal injury, and this was shown to adversely impact perioperative survival [3,5,6,7,8,9]. Advanced age, non-elective status, underlying liver disease, low body mass index (BMI) or body surface area (BSA), cardiopulmonary bypass duration, and performing five or more anastomoses, have been associated with the risk of re-exploration [3,5,6,10,11]. Delayed re-intervention with prolonged periods of hemodynamic instability and extensive use of blood products has been considered deleterious in this context, mandating a careful consideration of the indications for reopening and its optimal timing [12,13,14,15].

Recent studies have investigated the large-scale impact of re-exploration on clinical outcomes and costs in a large national survey, including factors related to hospital expertise, volume, and performance [16,17,18]. While confirming the detrimental effects of re-exploration in terms of postoperative mortality, morbidity and resource utilization, these Authors were unable to derive specific protocols or factors mitigating the risks and the negative outcomes of reoperation for bleeding [18]. Despite attempts made to investigate a decisional algorithm in those patients [19], many aspects of decision-making and correct timing for re-exploration still remain controversial, being left to the surgeon’s experience or judgment, especially in those borderline situations of hemodynamically stable patients with significant chest drain output [14,20,21,22].

We, therefore, sought to investigate in a propensity-matched analysis the impact of re-exploratory surgery on the short-term outcomes of elective CABG patients who experienced bleeding but were not in hemodynamically critical unstable conditions or required emergency reopening based on the hourly high chest tube output. The purpose was to determine the clinical outcomes of this “borderline population” according to the decision-making strategy adopted.

## 2. Materials and Methods

### 2.1. Study Design

A prospectively collected database of 8287 patients who underwent primary isolated elective CABG procedures between 2006 and 2021 at Blackpool Victoria Hospital was analyzed. Dyke universal definition of perioperative bleeding was used to identify patients who experienced severe or massive bleeding and were, therefore, candidate to re-exploration (“class 3”: Blood loss 1001–2000 mL in 12 h and “class 4”: Blood loss >2000 mL in 12 h) [23].

Patients with postoperative hemodynamical instability, as demonstrated by increasing inotropic support, need for mechanical assist devices, malignant arrhythmias (ventricular tachycardia, ventricular fibrillation, cardiac arrest), signs or imaging evidence of cardiac ischemia or tamponade were excluded. Moreover, patients with blood loss >200 mL per hour in the first 5 postoperative hours were excluded as surgical re-exploration was considered mandatory. Hemoglobin levels to testify to the significance of bleeding were not included in the analysis as records were deemed unreliable, considering that patients were clinically stable.

Rotational thromboelastometry (ROTEM) was performed in all patients with significant postoperative blood loss as per center routine. Patients with abnormal ROTEM or abnormal blood tests upon arrival in Intensive Care Unit (platelet count, prothrombin time, activated thromboplastin time and fibrinogen, activated clotting time, ACT) were excluded as the cause of bleeding might have been an underlying hematologic disorder or medical condition.

Therefore, a population of hemodynamically stable patients experiencing significant non-rapidly cumulating bleeding (>1000 mL of blood loss in 12 h) and normal point-of-care coagulation tests and platelet count was identified (N = 1642). In those patients, the consultant surgeon decided to re-explore or treat conservatively (i.e., wait-and-see with transfusional support only), according to his/her judgment. “Early re-exploration” indicates a revision for bleeding within 12 h from the index procedure, for bleeding that was considered “unacceptable” by the consultant surgeon despite not being rapidly-cumulating. Clinical outcomes according to the adopted decision-making strategy were compared using a propensity score matching approach.

Variables included in the analysis were perioperative mortality, myocardial infarction, acute renal injury, multiorgan failure, prolonged ventilation, respiratory complications, overall blood loss, periprocedural blood product use, length of stay in intensive care unit, and overall length of stay. The transfusion threshold was hemoglobin <8.5 g/dL, in the presence of signs or symptoms of reduced tissue oxygenation. Judgment on the clinical need for transfusion and hypo-perfusion was made on the basis of a comprehensive multimodal evaluation, including clinical, hemodynamic, and laboratory parameters (i.e., lactates) [24]. Criteria for blood component management and intraoperative anti-fibrinolytic agents were similar for all patients as per the center’s routine. Protamine was fully reversed (1–1.5 mg of protamine per 100 units of heparin) in the operating room to achieve an ACT similar to preoperative values, and no subsequent doses were administered if repeated ACT was normal (patients with abnormal ACT were excluded according to the protocol described above). Re-transfusion of shed mediastinal blood was not performed in patients undergoing CABG as per the center’s policy. 

The main aim of the study was to compare short-term postoperative outcomes and periprocedural blood product use in a subset of hemodynamically stable patients with significant perioperative blood loss on the basis of the decision-making strategy adopted (re-exploration vs. conservative management). IRB approval was obtained from Blackpool Victoria Hospital. Specific patient informed consent is not available considering the retrospective nature of this study.

Data acquisition and analysis were performed in compliance with protocols approved by the Ethical Committee of the Blackpool Victoria Hospital (Whinney Heys Road, Blackpool, Lancashire, FY38NR) (ethical approval number 2022-1902-B). Specific written informed consent was waived due to the retrospective nature of the study.

### 2.2. Statistical Analysis

Categorical variables are expressed as frequencies and percentages and compared using Chi-squared test or Fisher’s exact test, as appropriate. Continuous variables were checked for normality with Kolmogorov–Smirnov test. Normally distributed variables are shown as mean and standard deviation and compared with parametric tests (Student’s *t*-test). Not-normally distributed variables are presented as median and interquartile range and compared with non-parametric tests (Mann–Whitney U-test).

Patients’ cohorts were matched using a non-parsimonious propensity score matching algorithm using the 1-to-1 nearest neighbor method (considering a 0.20 caliper) without repetition, obtaining 251 unique, comparable couples. The logistic regression for estimation of the propensity score included: Age, sex, BSA, diabetes, smoking, hypertension, hypercholesterolemia, preoperative renal disease (dialysis), preoperative lung disease (corticosteroid treatment), preoperative neurological disease, preoperative extracardiac peripheral disease (EuroSCORE criteria), number of diseased vessels, left main disease, left ventricular ejection fraction (EuroSCORE criteria), number of distal anastomoses, on-pump surgery. This model was associated with a C-statistic of 0.835, and the diagnostic tests on the matched cohort showed adequate matching and meta-bias reduction (Rubin’s B 24.2%, Rubin’s R 0.89). Model details are presented in Appendix A.

Regression analysis was performed considering a binary, linear or ordinal variable with excess zeros as dependent variable, using respectively logistic, linear, or zero-inflated regression to estimate the effect of reoperation. Unadjusted (crude) regression was followed by propensity score-adjusted regression (with propensity score included as a covariate) and a regression for the matched cohort. A *p*-value less than 0.05 was considered statistically significant.

Statistical analysis was performed with STATA version 16 for Windows.

## 3. Results

From the entire database of 8287 elective patients undergoing CABG, 1642 (19.8%) were included in this analysis as experiencing significant but non-acute or rapidly cumulating bleeding while remaining in stable hemodynamical conditions. Patients’ demographics and preoperative characteristics of the entire population are reported in Table 1. A total of 252 patients (15.3%) were re-explored, and 1390 patients (84.7%) were treated conservatively. In all patients in the re-explored group, diffuse bleeding with no identifiable arterial surgical source was found.

Postoperative complications in the unmatched cohort are shown in Table 2 (left panel). In-hospital mortality was attributable to multiorgan failure (4 patients, 0.24%), sudden cardiac death (3 patients, 0.18%), respiratory complications (3 patients, 0.18%), sepsis (3 patients, 0.18%), gastrointestinal ischemia (1 patient, 0.06%), or massive stroke (1 patient, 0.06%).

After propensity score matching (Table 2, right panel), patients reoperated exhibited significantly higher total perioperative blood loss (1820 mL (1440–2300) vs. 1240 (1100–1500) *p* < 0.001). This was accompanied by a significantly higher overall blood product consumption, with more than 88% of the patients requiring more than 1 unit of blood and a median consumption of 3 units (2–4). A significantly higher proportion of patients in the reoperated group experienced prolonged ventilation and respiratory complications (110 (44.0) vs. 37 (14.8) *p* < 0.001). This finding was coupled with prolonged stay in intensive care unit (mean 1.61 ± 3.19 days vs. 1.28 ± 2.56, *p* < 0.001). Importantly, reoperated patients had a significantly increased risk of multiorgan failure after propensity match and a trend towards increased perioperative mortality (*p* = 0.057). No significantly increased rates of pericardial effusion requiring intervention (i.e., subxiphoid drainage) (*p* = 0.682) were found in this group. Finally, the overall length of stay in hospital was significantly higher in the reoperated group (7 (6–9) days vs. 6 (5–8) days, *p* = 0.004). Notably, no baseline differences were found in reoperated and conservatively treated patients, both before and after propensity score matching, confirming similar preoperative risk profiles and preoperative characteristics (Table 1).

Regression analysis (Table 3) confirmed that reoperation carries an increased risk of red packed cells (RPC) transfusion (coefficient 1.72, 95% CI 1.38–2.07, *p* = 0.001), increased blood loss (+546.5 mL, 95% CI 436.8–656.2 mL, *p* = 0.001), prolonged ventilation (odds ratio 4.52, 95% CI 2.94–6.94, *p* = 0.001), and multiorgan failure (odds ratio 4.59, 95% CI 1.37–15.42, *p* = 0.014). Notably, re-exploration was associated with longer postoperative Intensive Care Unit stay (+1.72 days, 95% CI 0.27–3.16, *p* = 0.020) and longer total postoperative hospitalization (+2.34 days, 95% CI 0.19–4.50, *p* = 0.033). Moreover, a significant trend towards increased in-hospital mortality was found in propensity score-adjusted regression in the unmatched population (odds ratio 3.12, *p* = 0.035), but this result was not significant in the matched cohort (*p* = 0.095).

## 4. Discussion

The present study investigated the outcomes of patients in stable hemodynamic conditions experiencing significant but non-critical postoperative bleeding after elective CABG. The main aim was to explore the decision-making conundrum regarding surgical re-exploration in borderline situations after CABG. Specifically, we identified patients with no critical bleeding, mandating immediate re-exploration but bleeding significantly while maintaining stable hemodynamic condition and normal point-of-care coagulation tests.

The main finding of this retrospective propensity-matched analysis is that re-exploration of hemodynamically stable patients with non-critical bleeding (i.e., >1000 mL in 12 h or not requiring emergency revision) is associated with (1) increased blood product consumption, (2) prolonged ventilation, (3) increased risk of multiorgan failure, and (4) increased length of stay in ICU and overall length of stay in hospital in comparison to matched patients managed conservatively. A trend towards increased perioperative mortality was also found in reoperated patients, but this did not reach statistical significance. Notably, the overall incidence of the mentioned borderline scenario of hemodynamically stable patients with significant bleeding almost reached 20% among patients undergoing CABG in this large single-center cohort.

Re-exploration has been associated with substantial morbidity and mortality. Previous studies have suggested that these patients have an in-hospital mortality over 3 times higher than patients not requiring re-exploration, as well as a greater in-hospital length of stay [2,13,25]. A systematic review and meta-analysis by Biancari et al. suggested that re-exploration for bleeding remained an independent predictor of immediate postoperative mortality [26].

Reasons underlying the poorer outcomes after re-exploration are multifactorial, and increased use of blood products, increased risk of acute renal injury and sepsis have been suggested as main determinants [12,13]. Potential additional explanations might be related to the higher preoperative risk profile of patients undergoing re-exploration. For these reasons, in the present study, a propensity-matched algorithm was used to flatten potential biases related to preoperative confounders.

Optimal timing in decision-making for re-exploration is controversial. Delayed re-exploration, defined as longer than 12 h after the procedure, was shown to be associated with increased transfusion requirements, increased mortality, and hospital stay [14,27]. In a large report from the NIS database, patients reoperated on the day of surgery had better outcomes and survival than those re-explored in the subsequent days [18]. Ruel et al. showed that re-exploration after the day of operation was associated with a 6.4-fold increase in the risk of death [7]. The prolonged hemodynamic instability, as well as the increased blood loss, were thought at the basis of these findings [27]. In support of this hypothesis, a study by Ranucci et al. concluded that the main determinant of morbidity and mortality for patients requiring surgical re-exploration was the amount of packed red cells transfused [12,13]. Furthermore, in a large clinical series re-exploration for bleeding was independently associated with a 3.5-fold increase in-hospital mortality when compared with conservative management [2]. However, it has been suggested that delaying the timing of re-exploration may represent a risk factor only when the delay creates the need for excessive use of allogeneic blood products [11].

In this study, we selected a borderline patient population with significant but not critical bleeding, normal coagulation profile, and stable hemodynamic conditions. In this cohort, all the re-explorations have been performed within 24 h from index surgery, and the focus of the study was to retrospectively determine outcomes on the basis of the surgical decision of re-exploration versus conservative management. Decision-making in these situations might be challenging, and there is no evidence guiding the best strategy to adopt, which is normally left to the surgeon’s experience. Besides the caveat of the retrospective nature of this study, we found that conservative management in these patients might provide better outcomes when compared to a strategy involving re-exploration. Explanations underlying these findings might relate to the fact that in patients able to maintain good hemodynamics, the bleeding culprit is rarely life-threatening or arterial and could potentially be self-contained or treated medically. Interestingly, the overall blood consumption (i.e., along the whole hospital stay) and the actual calculated overall blood loss were lower in the patients managed conservatively, allowing us to speculate that the reoperation itself (i.e., clots lysis and mediastinal wash-out) could further perturbate the established coagulation balance and eventually require further blood products. However, in this study, blood loss measurement could have been influenced by a number of reporting biases and cannot be considered a reliable variable to support this hypothesis. Instead, the surprisingly reduced blood product consumption in non-reoperated patients might explain the clinical outcome advantage of conservative management seen in this study, given the widely reported detrimental effects of blood derivatives after surgery [12,13,28,29]. Nevertheless, this study confirms the theoretical deleterious effect of re-exploration in terms of increased postoperative morbidity.

Importantly, resource utilization and expenditure associated with re-exploration are meaningful. In the present study, both ICU stay and overall hospital stay were significantly higher in re-explored matched cohort, and economic implications of conservative management in these stable borderline conditions warrant specific investigation. On the other side, the results of this study highlight the importance of defining protocols and management algorithms to optimize the care of these patients, both preoperatively and postoperatively. Efficacy of intraoperative checklists [30,31] and routine use of patient-targeted transfusion approaches [29,32], such as thromboelastography, have been reported, but several other aspects of decision-making in this context should be further investigated and improved, especially when depending on the clinical judgment or experience of the single surgeon. This study first attempted to explore this conundrum and provides initial, although retrospective, and hypothesis-generating data to orient decision-making in these borderline circumstances.

### Limitations

Among the limitations of this study, authors acknowledge the retrospective nature of the analysis and the presence of several hidden confounding biases which are known to affect observational studies [33]. The propensity match score analysis could not have adequately corrected other factors, such as surgeon’s personal experience or judgment, or other logistics issues associated with re-exploration.

The criterion for transfusion adopted in our center is liberal. Although this approach was equally adopted for all the patients in the study, and, therefore, no theoretical bias could have been introduced, it is difficult to predict if the present results would have been confirmed if a more restrictive criterion had been applied [34]. The impact of perioperative myocardial ischemia [35] might have influenced results, but no specific data were available for comparison in this study.

Another difficulty regards definitions. In this study, we selected a “borderline” population of patients experiencing significant bleeding, not acutely cumulating, or requiring emergency re-exploration, while maintaining stable hemodynamic conditions and normal coagulation parameters. In order to identify these patients, Dyke criteria [23] have been used, which universally codify bleeding after surgery, however, this classification does not include hemodynamic parameters or laboratory data. Additionally, hemoglobin levels are generally not used to testify to the significance of bleeding. The ROTEM charts and the drop in hemoglobin levels from ICU arrival to reoperation were not systematically recorded. Although we referred to a recognized classification for what concerns the bleeding measurement, to dichotomic variables to identify hemodynamic instability (use of inotropes, presence of pericardial effusion, occurrence of malignant arrhythmias), and to universal laboratory data to define patients’ coagulation profiles, we acknowledge the difficulty to achieve an actual standard definition for this “borderline” group of patients, which surely remains in a “grey zone” of the postoperative management. A larger sample size and multi-centric studies would be required to standardize this definition, however, despite preliminary, the study comprehensively captures this challenging and frequent scenario and provides insights into the outcomes of these patients on the basis of the surgical decision-making.

Details of preoperative medication regimens were not available in medical records, but patients with dual antiplatelet therapy or anticoagulants were considered by definition non-elective and were excluded from the study. Preoperative platelet mapping is not routinely performed in our center in elective cases. Preoperative acetylsalicylic acid administration was continued until the day of the surgery and was re-introduced after 6 h. Therefore, the preoperative and perioperative bleeding risk associated with medications could be considered homogeneous. The same reasoning applies to statins: They are known to be associated with a higher risk of hemorrhagic events [36], but different results have been found in cardiac surgery [37,38]. A more detailed understanding of their effect on postoperative bleeding in cardiac surgery is crucial, considering the unique role of cardiopulmonary bypass and inflammatory activation [39]. Considering the retrospective nature of the study and the modalities of data collection in the dataset, some variables cannot have a precise time-to-event collocation. Pre-reoperation and post-reoperation drainage output, as well as the exact time to reoperation, were not systematically recorded and, therefore, could not be used as exploratory or outcome variables. 

The tendency of the surgeon to see re-exploration for bleeding as a personal failure and avoid or delay reintervention as long as possible could have represented a significant unmeasurable confounder in this study. The re-explored population might include patients in which medical management within the first 24 h failed to control the bleeding within limits considered acceptable by the consultant surgeon. Clearly, the retrospective nature of the study exposes the effect of such unmeasured confounders, but the data regarding an overall reduced blood consumption and complications in matched patients conservatively managed invites us to reflect on the impact of re-exploratory surgery on CABG patients. A number of other variables, such as the actual incremental rate of bleeding, the type, location, and active clearance of the drains, could have assisted in understanding retrospectively the principles driving decision-making and also identify warning signs to guide management. Generally, the management of postoperative bleeding remains driven by the surgeon’s personal experience, and the results of this analysis should be interpreted considering these limitations and can only be considered hypothesis-generating.

## 5. Conclusions

The incidence in a high-volume center of patients experiencing non-acute, non-life-threatening but significant blood loss after elective CABG highlights the importance of this challenging circumstance and the need for evidence-based guidance in its perioperative management. Besides the caveat of the retrospective nature of this study and the inherent related biases and hidden confounding factors, conservative management in these borderline situations might be a viable option and might be advantageous in terms of reduction of postoperative morbidities and length of stay (Figure 1). Further investigations are warranted to confirm these findings.

## Figures and Tables

**Figure 1 jcm-12-03327-f001:**
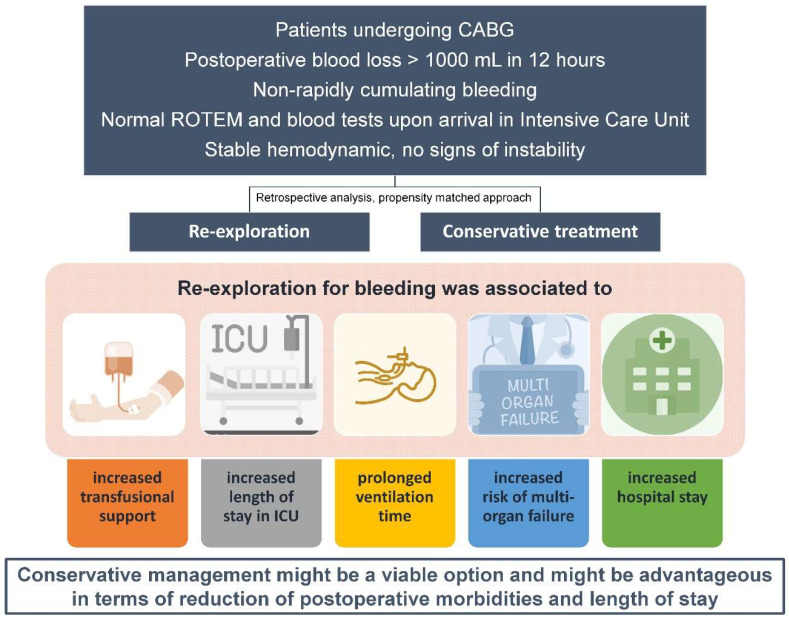
Main findings of the study.

**Table 1 jcm-12-03327-t001:** Patients’ demographics and preoperative characteristics.

	All Patients N = 1642	Unmatched Cohort, Not Reoperated N = 1390	Unmatched Cohort, Reoperated N = 252	*p* Value	Matched Cohort, Not Reoperated N = 251	Matched Cohort, Reoperated N = 251	*p* Value
Age	65.1 ± 9.6	64.9 ± 9.5	65.8 ± 10.1	0.226	66.1 ± 10.0	65.8 ± 10.1	0.667
Male sex	1397 (85.1)	1181 (84.9)	216 (85.7)	0.758	216 (86.1)	216 (86.1)	0.999
Body mass index (kg/m^2^)	26.7 ± 5.0	26.7 ± 5.2	26.7 ± 3.8	0.952	26.9 ± 3.8	26.7 ± 3.8	0.629
Body surface area (m^2^)	1.94 ± 0.30	1.94 ± 0.30	0.96 ± 0.29	0.322	1.97 ± 0.34	1.96 ± 0.29	0.879
Preoperative angina class							
0	204 (12.4)	169 (12.2)	35 (13.9)	0.397	30 (11.9)	35 (13.9)	0.237
1	247 (15.0)	203 (14.6)	44 (17.5)		37 (14.7)	44 (17.5)	
2	542 (33.0)	458 (32.9)	84 (33.3)		81 (32.2)	83 (33.1)	
3	442 (26.9)	377 (27.2)	65 (25.8)		62 (24.7)	65 (25.9)	
4	207 (12.6)	183 (13.2)	24 (9.5)		41 (16.3)	24 (9.6)	
Preoperative NYHA class							
0	230 (14.0)	191 (13.7)	39 (15.5)	0.065	43 (17.1)	39 (15.5)	0.158
1	634 (38.6)	524 (37.7)	110 (43.6)		84 (33.5)	109 (43.4)	
2	468 (28.5)	396 (28.5)	72 (28.6)		77 (30.7)	71 (28.3)	
3	277 (16.9)	249 (17.9)	28 (11.1)		40 (15.9)	28 (11.1)	
4	33 (2.0)	30 (2.2)	3 (1.2)		7 (2.8)	4 (1.6)	
Preoperative myocardial infarction							
no	795 (48.2)	683 (49.1)	112 (44.4)	0.337	120 (47.8)	112 (44.6)	0.620
<6 h	726 (44.2)	608 (43.7)	118 (46.8)		114 (45.4)	117 (46.6)	
6–24h	121 (7.4)	99 (7.1)	22 (8.7)		17 (6.8)	22 (8.7)	
Previous percutaneous coronary intervention	130 (7.9)	104 (7.5)	26 (10.3)	0.125	18 (7.2)	28 (11.1)	0.122
Diabetes							
no	1386 (84.4)	1174 (84.5)	212 (84.1)	0.844	218 (86.8)	211 (84.1)	0.567
diet treatment	63 (3.8)	51 (3.7)	12 (4.7)		14 (5.6)	12 (4.8)	
oral treatment	121 (7.4)	104 (7.5)	17 (6.7)		11 (4.4)	17 (6.8)	
insulin	72 (4.4)	61 (4.4)	11 (4.4)		8 (3.2)	11 (4.4)	
Smoking							
never	646 (39.3)	555 (39.9)	91 (36.1)	0.428	91 (36.2)	90 (35.8)	0.771
past	827 (50.3)	696 (50.1)	131 (52.0)		135 (53.8)	131 (52.2)	
current	169 (10.3)	139 (10.0)	30 (11.9)		25 (10.0)	30 (11.9)	
Hypertension	932 (56.8)	782 (56.2)	150 (59.5)	0.336	154 (61.3)	150 (59.8)	0.715
Hypercholesterolemia	909 (55.4)	783 (56.3)	126 (50.0)	0.063	126 (50.2)	126 (50.2)	0.999
Preoperative dialysis	18 (1.1)	17 (1.2)	1 (0.4)	0.246	2 (0.8)	1 (0.4)	0.563
Preoperative pulmonary disease	143 (8.7)	119 (8.6)	24 (9.5)	0.618	23 (9.2)	24 (9.5)	0.878
Preoperative stroke							
transient	79 (4.8)	65 (4.7)	14 (5.6)	0.727	16 (6.4)	14 (5.6)	0.932
stroke	26 (1.6)	23 (1.6)	3 (1.2)		3 (1.2)	3 (1.2)	
Preoperative extracardiac arteriopathy	246 (15.0)	212 (15.2)	34 (13.5)	0.471	36 (14.3)	33 (13.1)	0.697
Number of diseased vessels							
1	62 (3.8)	53 (3.8)	9 (3.6)	0.425	9 (3.6)	9 (3.6)	0.951
2	322 (19.6)	265 (19.1)	57 (22.6)		60 (23.9)	37 (22.7)	
3	1258 (76.6)	1072 (77.1)	186 (73.8)		182 (72.5)	185 (73.7)	
Left main disease	378 (23.0)	321 (23.1)	57 (22.6)	0.869	54 (21.5)	57 (22.7)	0.747
Left ventricular ejection fraction, category							
good (>50%)	1244 (75.8)	1050 (75.5)	194 (77.0)	0.723	191 (76.1)	193 (76.9)	0.975
fair (31–50%)	324 (19.7)	275 (19.8)	49 (19.4)		51 (20.3)	49 (19.5)	
poor (21–30%)	74 (4.5)	65 (4.7)	9 (3.6)		9 (3.6)	9 (3.6)	
Preoperative nitrates or heparin	52 (3.2)	40 (2.9)	12 (4.7)	0.116	9 (3.6)	12 (4.8)	0.504
Endoscopic vein harvesting	96 (5.9)	80 (5.7)	16 (6.3)	0.712	19 (7.6)	16 (6.4)	0.599
Skeletonized internal mammary artery	1642 (100)	1390 (100)	252 (100)	0.999	251 (100)	251 (100)	0.999
Number of distal anastomoses							
1	50 (3.0)	42 (3.0)	8 (3.2)	0.140	10 (4.0)	8 83.2)	0.895
2	310 (18.9)	248 (17.8)	62 (24.6)		63 (25.1)	62 (24.7)	
3	734 (44.7)	633 (45.5)	101 (40.1)		102 (40.6)	100 (39.8)	
4	465 (28.2)	395 (28.4)	70 (27.8)		62 (24.7)	70 (27.9)	
5	83 (5.0)	72 (5.2)	11 (4.4)		14 (5.6)	11 (4.4)	
On pump surgery	1374 (83.7)	1169 (84.1)	205 (81.3)	0.277	213 (84.8)	204 (81.3)	0.284
mean cardiopulmonary bypass time	88.5 ± 31.0	88.7 ± 30.7	87.6 ± 32.7	0.657	85.7 ± 31.8	87.7 ± 32.8	0.528
mean aortic cross clamp time	50.6 ± 22.6	50.3 ± 22.4	52.1 ± 24.0	0.287	49.6 ± 24.0	51.3 ± 23.9	0.473

**Table 2 jcm-12-03327-t002:** Postoperative complications.

	Unmatched Cohort Not Reoperated N = 1390	Unmatched Cohort Reoperated N = 252	*p* Value	Matched Cohort Not Reoperated N = 251	Matched Cohort Reoperated N = 251	*p* Value
RPC use(1+ units)	703 (50.6)	223 (88.5)	0.001	132 (52.6)	222 (88.4)	0.001
Transfused RPC units	1 (0–2)	3 (2–4)	0.001	1 (0–2)	3 (2–4)	0.001
Transfused FFP units	0 (0–2)	2 (0–4)	0.001	0 (0–2)	0 (2–4)	0.001
Transfused PLT units	0 (0–0)	4 (0–4)	0.001	0 (0–1)	4 (0–4)	0.001
Total blood loss	1220 (1080–1450)	1820 (1450–2290)	0.001	1240 (1100–1500)	1820 (1440–2300)	0.001
Stroke	16 (1.1)	2 (0.8)	0.616	4 (1.6)	2 (0.8)	0.411
Dialysis	12 (0.8)	7 (2.8)	0.009	3 (1.2)	7 (2.8)	0.201
Prolonged ventilation	157 (11.4)	111 (44.2)	0.001	37 (14.8)	110 (44.0)	0.001
Pulmonary complications	172 (12.4)	35 (13.9)	0.505	27 (10.7)	35 (13.9)	0.278
Gastrointestinal complications	26 (1.9)	6 (2.4)	0.590	5 (2.0)	6 (2.4)	0.760
Pericardial effusion *	161 (11.6)	32 (12.7)	0.613	29 (11.5)	32 (12.7)	0.682
CSAAKI	41 (2.9)	16 (6.3)	0.007	9 (3.6)	16 (6.4)	0.151
MOF	6 (0.4)	5 (2.0)	0.005	0 (0.0)	5 (2.0)	0.025
In-hospital mortality	9 (0.6)	6 (2.4)	0.008	1 (0.4)	6 (2.4)	0.057
In-hospital cardiac mortality	2 (0.1)	1 (0.4)	0.387	0 (0.0)	1 (0.4)	0.317
Intensive care unit stay, days	1 (1–1)	1 (1–1)	0.001	mean 1.28 ± 2.56 median 1 (1–1)	mean 1.61 ± 3.19 median 1 (1–1)	0.001
Postoperative hospital stay, days	6 (5–8)	7 (6–9)	0.001	6 (5–8)	7 (6–9)	0.004

RPC: Red packed cells, FFP: Fresh frozen plasma, PLT: Platelet concentrate, CSAAKI: Cardiac surgery-associated acute kidney injury (increase in serum creatinine by ≥0.3 mg/dL within 48 h after cardiac surgery or increase in serum creatinine of >1.5 fold of the baseline level), MOF: Multiorgan failure. * Pericardial effusion requiring subxiphoid drainage or pleuro-pericardial window.

**Table 3 jcm-12-03327-t003:** Effect of surgical re-exploration (vs. conservative management) on complications.

	Regression Method	Unadjusted Regression			Propensity Score Adjusted Regression			Matched Cohort Regression		
		Effect *	95% CI	*p* value	Effect *	95% CI	*p* Value	Effect *	95% CI	*p* Value
RPC use (1+ units)	logistic	7.51	5.03–11.22	0.001	7.72	5.15–11.56	0.001	6.90	4.35–10.92	0.001
RPC units	zero-inflated	1.97	1.72–2.23	0.001	2.01	1.75–2.28	0.001	1.72	1.38–2.07	0.001
FFP units	zero-inflated	1.48	1.05–1.90	0.044	1.53	1.10–1.95	0.048	1.37	0.75–1.98	0.156
PLT units	zero-inflated	1.55	1.29–1.81	0.001	1.51	1.24–1.77	0.001	1.39	0.74–2.05	0.220
Total blood loss	linear	590.2	523.8–656.7	0.001	596.5	529.4–663.7	0.001	546.5	436.8–656.2	0.001
Stroke	logistic	0.68	0.15–3.00	0.618	0.627	0.14–2.77	0.539	0.49	0.09–2.73	0.421
Dialysis	logistic	3.28	1.28–8.41	0.013	3.05	1.17–7.93	0.022	2.37	0.60–9.28	0.215
Prolonged ventilation	logistic	6.16	4.57–8.31	0.001	5.80	4.29–7.86	0.001	4.52	2.94–6.94	0.001
Pulmonary complications	logistic	1.14	0.77–1.69	0.505	1.06	0.71–1.58	0.765	1.34	0.78–2.29	0.279
Gastroint. complications	logistic	1.27	0.52–3.14	0.591	1.34	0.54–3.31	0.527	1.20	0.36–4.00	0.761
Pericardial effusion	logistic	1.15	0.49–2.67	0.627	1.19	0.52–2.72	0.657	1.21	0.39–3.95	0.734
CSAAKI	logistic	2.23	1.23–4.04	0.008	2.07	1.13–3.78	0.017	1.83	0.79–4.22	0.156
Multiorgan failure	logistic	4.66	1.41–15.41	0.011	4.59	1.37–15.42	0.014	collinearity	-	-
In-hospital mortality	logistic	3.74	1.32–10.61	0.013	3.12	1.08–8.99	0.035	6.12	0.73–51.22	0.095
Cardiac mortality	logistic	2.76	0.25–30.60	0.407	2.12	0.18–24.40	0.544	collinearity	-	-
Intensive care unit stay	linear	1.76	0.75–2.76	0.001	1.66	0.64–2.67	0.001	1.72	0.27–3.16	0.020
Postoperative stay	linear	2.75	1.00–4.50	0.002	2.16	0.42–3.91	0.015	2.34	0.19–4.50	0.033

* Effect: Odds ratio for logistic regression, coefficient for zero-inflated regression, and linear regression. See Table 2 for abbreviations.

## Data Availability

Data underlying this article will be shared on reasonable request.

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
