# Peer review of "Early Re-Exploration versus Conservative Management for Postoperative Bleeding in Stable Patients after Coronary Artery Bypass Grafting: A Propensity Matched Study"

_jcm, 2023, doi:10.3390/jcm12093327_

Round 1

Reviewer 1 Report (New Reviewer)

This paper is interesting as it addresses one of the most controversial patient encountered during real word practice. For such reason it would be interesting to add, if feasible, the reasons driving the consultants toward the decision for re-exploration; from the paper one of the reasons that can be inferred is the higher total blood loss but, as everyone working in cardiac surgery know, the reasons for bringing back to OR a patient hemodynamically stable and not "bleeding too much" can be different.

Author Response

Thank you for your comment and your appreciation. The reasons driving to re-exploration are difficult to be precisely defined, as patients with massive bleeding requiring mandatory re-exploration were excluded. Also, rapidly cumulating bleeding or unstable patients were excluded by study design. All the included patients had a significant but not rapidly cumulating bleeding (>1000 mL of blood loss in 12 hours, <200 mL per hour in the first 5 hours), and the decision for re-exploration was based on the personal evaluation: all patients with a bleeding which was considered “unacceptable” by the consultant surgeon. Those are patients often in the “gray zone” of the decisional process, and total blood loss was deemed unreliable to make the choice. We hope that we clarified our point.

Reviewer 2 Report (New Reviewer)

Post-operative bleeding is a severe medical issue, and assessing this problem is very important as it highly impacts the prognosis of patients. Nevertheless, the data from the current medical literature are concordant

with the subject treated in this article. Another article that should be included in the background session should be the "Elassal et al. article (1)". Adding this reference is significant as it develops an algorithm for selecting patients who primarily benefit from re-operation for bleeding complications.

Regarding the study design, some elements must be considered. As presented, patients referred to conservative management have a higher NYHA class III and IV percent as compared to the re-operated

patients. This feature can influence the final result regarding the increased stay in the ICU and blood consumption. Also, hypercholesterolemia is more frequent in conservative management patients. Given the data regarding the statin association with bleeding complications – especially intracerebral haemorrhage - (2) data regarding statin dose in the two groups should be provided.

Meanwhile, assessing the median time until the bleed to re-operation is vital. This data should also be provided. The data from some of the excluded patients   – due to abnormal coagulation tests or abnormal ROTEM – can still bring precious information, as these individuals may need longer antiplatelet therapy due to the severity of the ischemic disease. Even if these tests can influence the results, these patients should have been included in the study, with a separate statistical analysis performed for this subgroup.

The statistical assessment is properly designed, and the data is concise and comprehensively presented.

The discussion session is very well structured; it includes all data that should be considered.

The study limitations bring a critical aspect, which is presented, namely the surgeons' experience, as this factor can influence both the bleeding complications and their management.

The manuscript brings essential data regarding managing post-procedural bleeding complications after CABG. The statistical data are properly presented, but some changes should

be made in the study design, and some supplementary data need to be provided.

1. Elassal, A.A., Al-Ebrahim, K.E., Debis, R.S. et al. Re-exploration for bleeding after cardiac surgery: revaluation of urgency and factors promoting low rate. J Cardiothorac Surg 16, 166 (2021).

https://doi.org/10.1186/s13019-021-01545-4

2. Association of Long-term Statin Use With the Risk of Intracerebral Hemorrhage. A Danish
Nationwide Case-Control Study. Daniel Albjerg Rudolph, Stine Munk Hald, Luis Alberto García
Rodríguez, Sören Möller, Jesper Hallas, Larry B. Goldstein, David Gaist. Neurology Aug 2022, 99
(7) e711-e719; DOI: 10.1212/WNL.0000000000200713

Author Response

Reviewer 2

Post-operative bleeding is a severe medical issue, and assessing this problem is very important as it highly impacts the prognosis of patients. Nevertheless, the data from the current medical literature are concordant with the subject treated in this article. Another article that should be included in the background session should be the "Elassal et al. article (1)". Adding this reference is significant as it develops an algorithm for selecting patients who primarily benefit from re-operation for bleeding complications.

Thank you for your comment and appreciation. The reference has been included as suggested.

Regarding the study design, some elements must be considered. As presented, patients referred to conservative management have a higher NYHA class III and IV percent as compared to the re-operated patients. This feature can influence the final result regarding the increased stay in the ICU and blood consumption. Also, hypercholesterolemia is more frequent in conservative management patients. Given the data regarding the statin association with bleeding complications – especially intracerebral haemorrhage - (2) data regarding statin dose in the two groups should be provided.

Thank you for your comment. Unfortunately, we do not have data about statin dose. However, differences in NYHA class and statin use can be found only in the unmatched population (i.e. before propensity score). Data about matched population are similar. Discussion has been expanded considering data about statins.

Meanwhile, assessing the median time until the bleed to re-operation is vital. This data should also be provided. The data from some of the excluded patients   – due to abnormal coagulation tests or abnormal ROTEM – can still bring precious information, as these individuals may need longer antiplatelet therapy due to the severity of the ischemic disease. Even if these tests can influence the results, these patients should have been included in the study, with a separate statistical analysis performed for this subgroup.

As stated in limitations, “Pre-reoperation and post-reoperation drainage output, as well as the exact time to reoperation, were not systematically recorded and therefore could not be used as exploratory or outcome variables.” Therefore, median time to reoperation is not available. Patients with abnormal coagulation tests were excluded by study design and data were not collected. Those points can be crucial to investigate the outcomes in another tailored study. Thank you for your suggestion.

The statistical assessment is properly designed, and the data is concise and comprehensively presented.

Thank you for your comment and appreciation.

The discussion session is very well structured; it includes all data that should be considered.

Thank you for your comment and appreciation.

The study limitations bring a critical aspect, which is presented, namely the surgeons' experience, as this factor can influence both the bleeding complications and their management.

Thank you for your comment and appreciation.

The manuscript brings essential data regarding managing post-procedural bleeding complications after CABG. The statistical data are properly presented, but some changes should be made in the study design, and some supplementary data need to be provided.

Thank you for your comment and appreciation. We hope that the all your points have been answered adequately.

Reviewer 3 Report (New Reviewer)

A somewhat interesting study related to the decision to re-explore patients with a defined amount of chest tube drainage after cardiac surgery.  It is a well written and somewhat novel approach to the question.  Why did you decide to study patients with normal coagulation testing.  I would have guessed after caring for these patients in the ICU a number of years that patients with normal coags would have been more likely to have a surgical bleed.  Acute more aggressive bleeding from a surgical bleed can cause a coagulopathy as well.  Maybe the fact that coags were normal was a good sign that the bleeding was not significant and that you could be more conservative. I think the overall result of this study was that if there is a question whether to re-explore or not, you should wait. Certainly the need for a second operation, transporting the patient to the operating room and back, will increase mortality which is not new information

Author Response

Thank you for your comment and your appreciation. We decided to include patients with normal coagulation tests because otherwise alterations in coagulations would have explained the bleeding, and correction of coagulation factors was by far the most important thing to do. Surgical bleeding is common in patients with normal coagulation tests, but in most cases no source of bleeding can be found in case of profuse but not rapidly-cumulating bleeding. It is obvious that massive bleeding with hemodynamic instability is from a major source (e.g. aortic cannulation site), but in diffuse bleeding the so-called “negative re-exploration” are common and surgical re-exploration can be therapeutic (DOI: 10.1016/s0003-4975(98)00078-2. On the other hand, sticking to a purely conservative approach has been shown to be detrimental (DOI: 10.1016/j.athoracsur.2022.07.037), and this topic is debated since the late ‘90s. To summarize our paper, waiting can be a good option but only in selected cases. We made minor changes to the manuscript.

This manuscript is a resubmission of an earlier submission. The following is a list of the peer review reports and author responses from that submission.

Round 1

Reviewer 1 Report

Dr. Spadaccio et al. conducted a propensity-score matched study including 1642 patients who underwent isolated CABG and experienced "non acute, hemodynamically stable post operative bleeding. Of those, 252 patients were re-explored and 1390 patients were managed conservatively. Interestingly, among 252 patients who were reexplored, no identifiable arterial sources were found. After propensity score matching, 252 pairs of patients including, which showed that patients who were re-operated had significantly higher blood loss, blood product consumption, and associated with prolonged intubation, increased risk of MOF, and length of stay. The topic is very relevant to patients management and important. However, even after the propensity score matching, re-exploration group has significantly higher blood loss. Therefore, my concern is that favorable outcomes for conservative group might have been driven by the less blood loss rather than the non op management itself. I don't think these results were useful guide when to explore for post cabg patients until authors resolved this issue.  

Author Response

Authors are thankful to reviewer for his/her comments. In this study, the propensity match analysis was performed to equalize the 2 populations in terms of risk profile, thus avoiding potential confounding bias that could make one group more prone to bleeding or negative outcomes. The logistic regression for estimation of the propensity score included: age, sex, BSA, diabetes, smoking, hypertension, hypercholesterolemia, preoperative renal disease (dialysis), preoperative lung disease (corticosteroid treatment), preoperative neurological disease, preoperative extracardiac peripheral disease (EuroSCORE criteria), number of diseased vessels, left main disease, left ventricular ejection fraction (EuroSCORE criteria), number of distal anastomoses, on-pump surgery. This model was associated with a C-statistic of 0.835, and the diagnostic tests on the matched cohort showed adequate matching and meta-bias reduction (Rubin’s B 24.2%, Rubin’s R 0.89). We appreciate your concern, however, considering the risk profile of the 2 populations is similar, we believe that the finding of the difference in blood loss is likely a consequence of the management (conservative vs reoperative) rather than a confounder driving the results. In fact, we aimed at demonstrating difference in outcomes as bleeding, stroke, death infection etc. Nevertheless, when adding to the PSM model the postoperative bleeding, results did not change, thus suggesting the findings are reliable.

Reviewer 2 Report

The authors present a retrospective study on early re-exploration vs conservative management for postoperative bleeding in stable patients after CABG: a propensity matched study. However, I have several concerns regarding the scientific value of this manuscript:

1.       How did the authors define “early” re-exploration in their study?

2.       Please remove “long-term survival” and use perioperative survival.

3.       What is exactly and how the authors define “significant but not acutely cumulating chest drain output”? Pls remove “profuse non-acute bleeding” due to the fact that ANY bleeding is acute. <200 mL per hour in the first 5 hours is still indication for re-exploration. How do the authors compare this with the well-known rule 150+150 or 100+100+100, pls discuss

I also suggest removing the term “significant non-acute bleeding” from the whole text.

4.       How the authors define “borderline population”? Please re-phrase.

5.       How did the authors define “normal platelet and coagulation test” in 1642 pts with (>1000 mL of drainage in 12h. Did they perform ROTEM in every patient? Can the authors provide comparative table of the results in both groups as this is crucial for your comparison between the groups?

6.       They state:” After PSM, reoperated patients exhibited significantly higher overall blood product consumption” If the group of patients that were conservatively treated, which means with blood products, had less blood transfusion, than you did not match the patients well. These pts were treated preferably with blood transfusion; therefore, it is impossible to have less products given per patient in comparison with surgically treated patient. Matching patients is rather complicated here and this is my main concern regarding your study as it can bias your results and mislead the conclusions. Both groups were matched based on wrong data, which are preoperative data, as there was no matching based on coagulation status, chest drainage, inotropic support and Hb drop. Reoperated group had significantly higher drain output which makes your groups non-comparable and conclusion invalid. Main reason why the patient in reoperated groups had surgery and more blood products is because they had more bleeding (“acute” bleeding), simple as that.

7.       The authors should report the numbers in the result section of the abstract.

8.       The conclusion seems misleading and consists of overstatements due to the previously mentioned comments.

9.       How did the authors define in these patients that there was no increase in inotropic support?

10.   The authors state: “Hemoglobin levels to testify the significance of bleeding were not included in the analysis as records were deemed unreliable considering that patients were clinically stable”. Can the authors explain this and provide the average drop in Hb in both groups of patients as this is clinically important?

11.   Did the authors compare perioperative MI when discussing and comparing mortality of both groups? Pls refer to the recent important study of post CABG mortality (Robinson N et al. Postcardiac surgery myocardial ischemia: Why, when, and how to intervene. J Thorac Cardiovasc Surg. 2021 Jul 10:S0022-5223(21)01031-X.).

12.   Discussion and limitations need to be corrected accordingly and conclusion re-written as it is not justified with the results in the current form.

13.   Pls remove the discussion about the cost analysis as this was not the part of your study.

14.   Pls remove Fig 1 as it does not add anything new to this article and is biased.

15.   The whole text needs to be improved for English language and grammar. Pls correct some weird wording and use standard terms or be more precise: “time on cardiopulmonary bypass”, “Older age”, “oozing profusely”, “elevated incidence”, “the management of postoperative bleeding remains based on expert opinion”,...

Author Response

The authors present a retrospective study on early re-exploration vs conservative management for postoperative bleeding in stable patients after CABG: a propensity matched study. However, I have several concerns regarding the scientific value of this manuscript:

  1. How did the authors define “early” re-exploration in their study?

Thanks for your comment. Early re-exploration indicates reoperated within 12 hours. A note has been made to change the manuscript accordingly

  1. Please remove “long-term survival” and use perioperative survival.

Thanks for your comment, the term has been changed.

  1. What is exactly and how the authors define “significant but not acutely cumulating chest drain output”? Pls remove “profuse non-acute bleeding” due to the fact that ANY bleeding is acute. <200 mL per hour in the first 5 hours is still indication for re-exploration. How do the authors compare this with the well-known rule 150+150 or 100+100+100, pls discuss. I also suggest removing the term “significant non-acute bleeding” from the whole text.

Thanks for your comment. We selected a population of hemodynamically stable patients experiencing significant “oozing” but not bleeding with rapid accumulation in the  drains. In our practice we utilize a protocol according to Dyke criteria. Bleeding >200 ml per hour in the first 5 hours is considered a criterion for reopening. As you know, in the clinical practice many factors influence the decision on reopening. In some instances, the surgeon, knowing the intraoperative coagulation status and the potential surgical bleeding points, would allow for a more conservative approach or a rapid re exploration. However, there is a population of patients that would constantly oozing with normal coagulation profile, but not really meeting the criteria to warrant an emergency re-exploration. This is the type of challenging population (i.e. “border line”) that we wanted to target. We removed the terms “profuse” and “significant non-acute bleeding” as suggested.

  1. How the authors define “borderline population”? Please re-phrase.

As explained in comment 3, we targeted a population of patients that would constantly oozing with normal coagulation profile after surgery, but not really meeting the criteria to warrant an emergency re-exploration. In this scenario, decision making could be challenging as balancing the risk of reoperation with the risk of leaving a potential bleeding unexplored. Given the characteristics and the timing, this type of bleeding is most probably related to the venous system. However, this population “border line” for reopening can be challenging to address. For this reason we performed this study.

  1. How did the authors define “normal platelet and coagulation test” in 1642 pts with (>1000 mL of drainage in 12h. Did they perform ROTEM in every patient? Can the authors provide comparative table of the results in both groups as this is crucial for your comparison between the groups?

As described in methods, “Rotational thromboelastometry (ROTEM) was performed in all patients with significant postoperative blood loss as per center routine; patients with abnormal ROTEM or abnormal blood tests upon arrival in Intensive Care Unit (platelet count, prothrombin time, activated thromboplastin time and fibrinogen, activated clotting time, ACT) were excluded as the cause of bleeding might have been an underlying hematologic disorder or medical condition.”. ROTEM were evaluated by experienced physicians, but ROTEM charts were not systematically recorded and are not available.

  1. They state:” After PSM, reoperated patients exhibited significantly higher overall blood product consumption” If the group of patients that were conservatively treated, which means with blood products, had less blood transfusion, than you did not match the patients well. These pts were treated preferably with blood transfusion; therefore, it is impossible to have less products given per patient in comparison with surgically treated patient. Matching patients is rather complicated here and this is my main concern regarding your study as it can bias your results and mislead the conclusions. Both groups were matched based on wrong data, which are preoperative data, as there was no matching based on coagulation status, chest drainage, inotropic support and Hb drop. Reoperated group had significantly higher drain output which makes your groups non-comparable and conclusion invalid. Main reason why the patient in reoperated groups had surgery and more blood products is because they had more bleeding (“acute” bleeding), simple as that.

Authors are thankful to reviewer for his/her comments. We agree with the reviewer that matching was complicated, but our approach tried to flatten bias to provide reliable results. In this study, the propensity match analysis was performed to equalize the 2 populations in terms of risk profile, thus avoiding potential confounding bias that could make one group more prone to bleeding or negative outcomes. As stated, “Pre-reoperation and post-reoperation drainage output, as well as the exact time to reoperation, were not systematically recorded and therefore could not be used as exploratory or outcome variables.”. Postoperative (i.e. post-re-exploration for bleeding) variables, such as the amount in the chest tube or Hb levels, cannot be adequately evaluated in terms of timing and their use in the PSM method would have increased the bias. We included only “pre-re-exploration” variables to simulate an evaluation at the of the decision between reoperation and conservative management. The logistic regression for estimation of the propensity score included: age, sex, BSA, diabetes, smoking, hypertension, hypercholesterolemia, preoperative renal disease (dialysis), preoperative lung disease (corticosteroid treatment), preoperative neurological disease, preoperative extracardiac peripheral disease (EuroSCORE criteria), number of diseased vessels, left main disease, left ventricular ejection fraction (EuroSCORE criteria), number of distal anastomoses, on-pump surgery. This model was associated with a C-statistic of 0.835, and the diagnostic tests on the matched cohort showed adequate matching and meta-bias reduction (Rubin’s B 24.2%, Rubin’s R 0.89). We appreciate your concern, however, considering the risk profile of the 2 populations is similar, we believe that the finding of the difference in blood loss is likely a consequence of the management (conservative vs reoperative) rather than a confounder driving the results. In fact, we aimed at demonstrating difference in outcomes as bleeding, stroke, death infection etc. Nevertheless, when adding to the PSM model the postoperative bleeding, results did not change, thus suggesting the findings are reliable.

  1. The authors should report the numbers in the result section of the abstract.

Thank you for your comment. Numerical results have been included in the abstract.

  1. The conclusion seems misleading and consists of overstatements due to the previously mentioned comments.

Considering the retrospective nature of this study, our findings remain speculative, and hypothesis-generating as causal relationship requires different study design. This has been extensively discussed in the manuscript. In this study, the propensity match analysis was performed to equalize the 2 populations in terms of risk profile, thus avoiding potential confounding bias that could make one group more prone to bleeding or negative outcomes. We appreciate your concern, however, considering the risk profile of the 2 populations is similar, we believe that the finding of the difference in blood loss is likely a consequence of the management (conservative vs reoperative) rather than a confounder driving the results. Nevertheless, when adding to the PSM model the postoperative bleeding, results did not change, thus suggesting the findings are reliable.

  1. How did the authors define in these patients that there was no increase in inotropic support?

Increasing inotropic support is considered a sign of hemodynamic instability and those patients are referred for urgent re-exploration. Therefore, “increasing inotropic support” was an exclusion criteria of the study and was always reported in the clinical records.

  1. The authors state: “Hemoglobin levels to testify the significance of bleeding were not included in the analysis as records were deemed unreliable considering that patients were clinically stable”. Can the authors explain this and provide the average drop in Hb in both groups of patients as this is clinically important?

In the early postoperative hours, hemodilution from cardiopulmonary bypass and the loss of whole blood make hemoglobin level unreliable to quantify postoperative bleeding. An average drop in Hb is unfortunately not available as those data were not collected. This point was emphasized in the limitations of the study.

  1. Did the authors compare perioperative MI when discussing and comparing mortality of both groups? Pls refer to the recent important study of post CABG mortality (Robinson N et al. Postcardiac surgery myocardial ischemia: Why, when, and how to intervene. J Thorac Cardiovasc Surg. 2021 Jul 10:S0022-5223(21)01031-X.).

Thank you for your comment. Intraoperative graft verification with flow assessment was routinely introduced in our center in recent years. Previous data about perioperative MI were based on postprocedural CKMB and TnI levels. Perioperative MI were not recorded in the database as the primary aim of the study was a comparison between 2 surgical approached in case of postoperative bleeding. This limitation and the relative implications have been described and the reference has been included.

  1. Discussion and limitations need to be corrected accordingly and conclusion re-written as it is not justified with the results in the current form.

In this study, the propensity match analysis was performed to equalize the 2 populations in terms of risk profile, thus avoiding potential confounding bias that could make one group more prone to bleeding or negative outcomes. The logistic regression for estimation of the propensity score included: age, sex, BSA, diabetes, smoking, hypertension, hypercholesterolemia, preoperative renal disease (dialysis), preoperative lung disease (corticosteroid treatment), preoperative neurological disease, preoperative extracardiac peripheral disease (EuroSCORE criteria), number of diseased vessels, left main disease, left ventricular ejection fraction (EuroSCORE criteria), number of distal anastomoses, on-pump surgery. This model was associated with a C-statistic of 0.835, and the diagnostic tests on the matched cohort showed adequate matching and meta-bias reduction (Rubin’s B 24.2%, Rubin’s R 0.89). We appreciate your concern, however, considering the risk profile of the 2 populations is similar, we believe that the finding of the difference in blood loss is likely a consequence of the management (conservative vs reoperative) rather than a confounder driving the results. In fact, we aimed at demonstrating difference in outcomes as bleeding, stroke, death infection etc. Nevertheless, when adding to the PSM model the postoperative bleeding, results did not change, thus suggesting the findings are reliable. Authors are thankful to the reviewer for his/her comments, limitations have been extensively discussed as well as the rationale of the study.

  1. Pls remove the discussion about the cost analysis as this was not the part of your study.

Thank you for your comment. The sentence has been rephrased to describe a potential implication for future studies.

  1. Pls remove Fig 1 as it does not add anything new to this article and is biased.

Figure 1 is intended as a “graphical abstract” summarizing the main findings of the article. All the limitations of the article have been extensively summarized and we believe that this figure might help readers in understanding the point of the manuscript.

  1. The whole text needs to be improved for English language and grammar. Pls correct some weird wording and use standard terms or be more precise: “time on cardiopulmonary bypass”, “Older age”, “oozing profusely”, “elevated incidence”, “the management of postoperative bleeding remains based on expert opinion”,...

Thank you for your comment. Language has been revised throughout the manuscript.

Round 2

Reviewer 1 Report

I'm not convinced the authors' opinion that difference in blood loss is likely from the consequence of the management just because the two groups have similar risk profiles. Authors need to present the data suggesting that reexploration is the cause of the difference in blood loss, rather than patients who had more blood loss ended up undergoing re-exploration and having more comorbidities.  

Reviewer 2 Report

The authors have addressed some of the previous comments; however, they have only performed minor revision and did not address important suggestions.

1. I do not agree that early re-exploration should refer to patients re-operated within 12h for a simple reason. If the patient had CABG in the morning and was re-explored overnight, it does not fall into your criteria but surely is early bleeding and early re-exploration.

2. I still have concerns about your "borderline" population as these criteria have not been established in routine practice. Patients in your cohort have been largely selected and concept of patients "significantly oozing but not bleeding with rapid accumulation" is wrong and misleading, leads to several biases and leads to wrong conclusions.

3. The authors state that they performed ROTEM in all these patients but did not provide data for a single patient. Therefore, their comparison does not seem reliable.

4. ROTEM data are more important for comparison of bleeding in these 2 groups than insignificant data such as diabetes, HLP, and smoking.

5. PSM analysis and comparison of these 2 groups of largely selected patients is meaningless. The authors did not consider the most important surgical factors such as Hb and coagulation status. On the other hand, they performed statistical analysis including irrelevant data such as diabetes, HLP, and smoking. PSM is severely biased and misleading in real clinical scenarios.

6. Fig 1 does not add anything to the readership of the manuscript and is irrelevant.